# Trends of the Florida manatee (*Trichechus manatus latirostris*) rehabilitation admissions 1991-2017

Ray L. Ball[1☯¤]*, Markuu Malmi[2☯], Janice Zgibor[2☯]

1 Department of Medical Sciences, ZooTampa at Lowry Park, Tampa, FL, United States of America,
2 College of Public Health, University of South Florida, Tampa, FL, United States of America

☯ These authors contributed equally to this work.
¤ Current address: Department of Biology, Eckerd College, St. Petersburg, Florida, United States of America
* ballrl@eckerd.edu

**Data Availability Statement:** All relevant data are within the paper and its Supporting Information files.

## Abstract

A retrospective study of admission data of 401 West Indian manatees (*Trichechus manatus latirostris*) presented to the David A. Straz Jr. Manatee Critical Care Center at ZooTampa at Lowry Park (ZooTampa) for rehabilitation from August 1991 through October 2017. Causes of admittance, location of rescue, gender, and age class were all recorded for each manatee admitted. Admittance categories as defined by the Florida Fish and Wildlife Conservation Commission (FWC) included watercraft collisions, natural causes, entanglement, entrapment, orphaned calves, captive born, mothers of rescued calves, calves of rescued mothers, human, and other. The admitted population was primarily from the southwest and northwest coasts and related waterways of Florida. The gender difference was relatively equivocal (54% female) while the adults comprised 79% of the admissions. The overall total admissions increased steadily over the study period as did the admissions for each individual categories of admission. Watercraft collisions and natural causes combined were 71% of all admissions for the entire study period and are the dominant causes of admission. Watercraft collisions are more likely to occur during May through October, whereas natural causes of admittance are more likely to occur between December and March. Rehabilitated manatees may reduce overall manatee mortality and can provide insight into population-based health concerns if evaluated appropriately. Future efforts can incorporate physical examination findings, hematology, biochemistry profiles, and ancillary diagnostic testing to continue to improve the individual welfare of this marine mammal in its natural range. Admissions data could also potentially serve the wider conservation and recovery efforts if it is proven that the data obtained is at least as informative as that obtained by the carcass salvage program. Limited conservation resources could then be re-directed as new challenges arise with the expanding population and potentially expanding range of this species.

**Funding:** The author(s) received no specific funding for this work.

**Competing interests:** The authors have declared that no competing interests exist.

## Introduction

The West Indian manatee is comprised of two subspecies; the Antillean (*Trichechus manatus manatus)* and the Florida manatee (*Trichechus manatus latirostris)*. Watercraft-related deaths and potential loss of warm water refuges are the primary threat to manatee populations [1]. In addition to these threats, drowning due to canal locks and flood gates, entanglement in fishing gear, cold exposure, red tide outbreaks, and habitat loss have all contributed manatee morbidity and mortality and necessitated manatee rescues. The Marine Mammal Protection Act of 1972 [2], Endangered Species Act of 1973 [3], and the Florida Manatee Sanctuary Act of 1978 [4] prohibit any killing, capture, or inhumane harassment of manatees. The West Indian manatee Recovery Plan [5] was implemented in March 1980 and provided a framework to provide protection of this species. As a result of actions such as enforcement of manatee speed zones in waterways, providing manatee sanctuaries, restoration of aquatic vegetation, and education on manatee conservation, the population of the Florida manatee steadily climbed. In 1991, there were an estimated 1,267 Florida manatees, whereas in early 2017 the population was estimated at 6,620 [6]. As of March 30, 2017, the endangered status of the West Indian manatee has been changed to "Threatened" by the United States Fish and Wildlife Service (FWS) under the Endangered Species Act [7]. This change does not affect other federal and state protections afforded manatees.

There are four manatee management units in Florida which include the Upper St. John's River with an estimated 4% of the population, the Atlantic Coast with 46% of the population, Southwest Florida with 38% of the population, and Northwest Florida with 12% of the population. ZooTampa, Miami Sea Aquarium, Sea World Orlando, and Jacksonville Zoo, are federally permitted critical care facilities for manatee rehabilitation. Identifying the effectiveness of the rehabilitation efforts is essential to all rehabilitation programs and the foundation for improved release rates, enhanced welfare, and optimal use of resources. This analysis of manatee admittance data at ZooTampa seeks to identify trends of admission rates and locations of rescue stratified by cause of admittance, age, and gender between January 1991 and October 2017. Preliminary analysis of the admittance data may also be useful to predict trends within the population, especially if the rehabilitation data on individual admissions can be summarized and parallels the trends seen in the wild population.

## Materials and methods

ZooTampa, formerly Lowry Park Zoo, has been rehabilitating injured and distressed Florida manatees since August 1991. Manatees that have obvious injuries or are exhibiting abnormal behaviors such as unusual buoyancy and lethargy are typically reported to FWC. Biologists are then dispatched to investigate and determine the need for intervention. Manatees determined to need medical assistance are captured by FWC teams and transported to one of the four federally permitted rehabilitation centers in Florida. Because qualified, practicing veterinarians are not typically on the rescue site nor are they transporting manatees, no medical support is provided until arrival at a critical care facility. On arrival at ZooTampa, manatees are triaged, baseline data is collected including blood sampling, lifesaving procedures are performed if indicated, and the vast majority of the manatees are hospitalized. The ZooTampa manatee medical database from August of 1991 through October 2017 was reviewed and included a total of 429 manatees. Data collection ended in October 2017 when the manatee care center underwent major renovations. Twenty-eight manatees were excluded due to incomplete data. Study variables included gender, age class, cause of admittance, and location of rescue. Mortality, release, and days in hospital were also collected but are not reported here as they are related to outcomes. Admittance categories as well as causes of death are defined by the Florida Fish and Wildlife Conservation Commission and include watercraft collisions, natural causes (cold

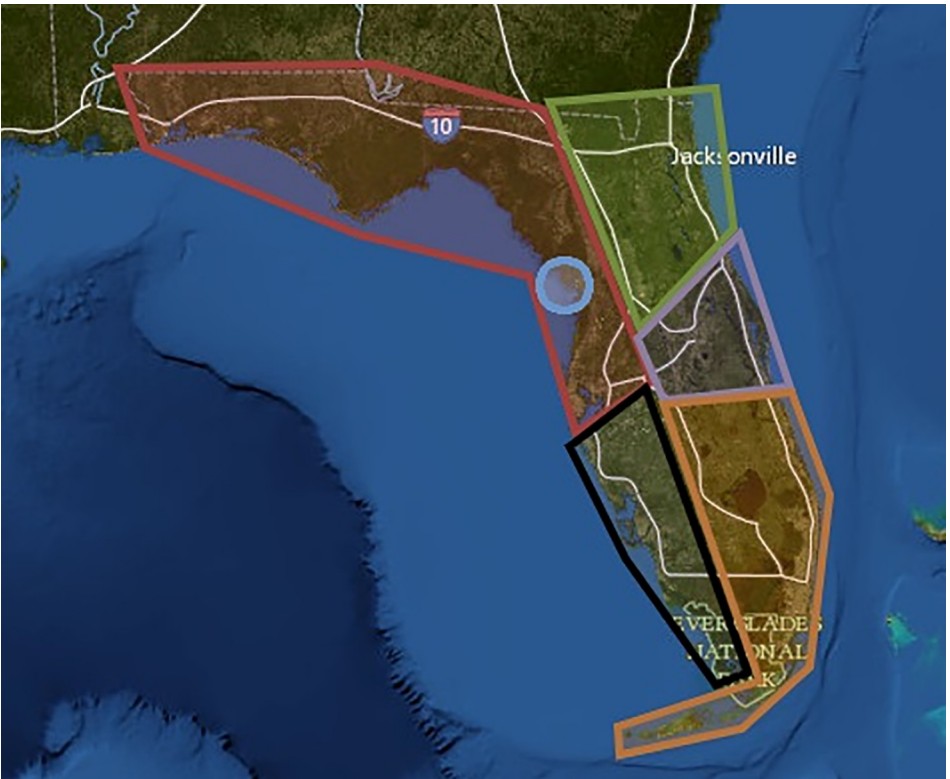

**Fig 1. Florida Fish and Wildlife Commission rescue and carcass salvage program map of Florida.** Red outline–Northwest (NW), Green outline–Northeast (NE), Purple outline–East Coast (EC), Orange outline–Southeast (SE), Black outline–Southwest (SW). Blue Circle–Crystal River (CR). Map provided by http://viewer.nationalmap.gov/viewer/.

stress, brevetoxicosis, anything non-human related), other human causes (entanglement, entrapment, captive born, or other causes) and orphaned calves. Mothers of rescued calves and calves of rescued mothers were included in the appropriate category of the manatee requiring rehabilitation. Rehabilitated orphaned manatees must obtain 200cm before being qualified for release. Straight length criteria for manatees is utilized to categorize various life stages of manatees by the biologist. Calves are classified as < 235cm, sub-adults from 235 to 265cm, and adults > 265cm [8]. A criteria of 200cm was chosen for this study as that straight length is a determinant for both rescue and release criteria. Orphaned calves were defined for this study as calves less than 200cm straight length. Any isolated manatee less than 200cm is considered a dependent calf and will be rescued if possible. Calves undergoing rehabilitation must be 200cm before being considered for release.

FWC manages the manatee rescue and carcass salvage program and divides the Florida into 5 sections (Fig 1); Northeast (NE), East Coast (EC), Southeast (SE), Southwest (SW), and

**Table 1. Manatee admissions data at ZooTampa August 1991 to October 2017.**

|  | Male | Female | Total | % Population of total |
|---|---|---|---|---|
| Adult | 138 | 177 | 315 | 79 |
| Calf | 46 | 40 | 86 | 21 |
| Total | 184 | 217 | 401 | |
| % Population of total | 46 | 54 | | |

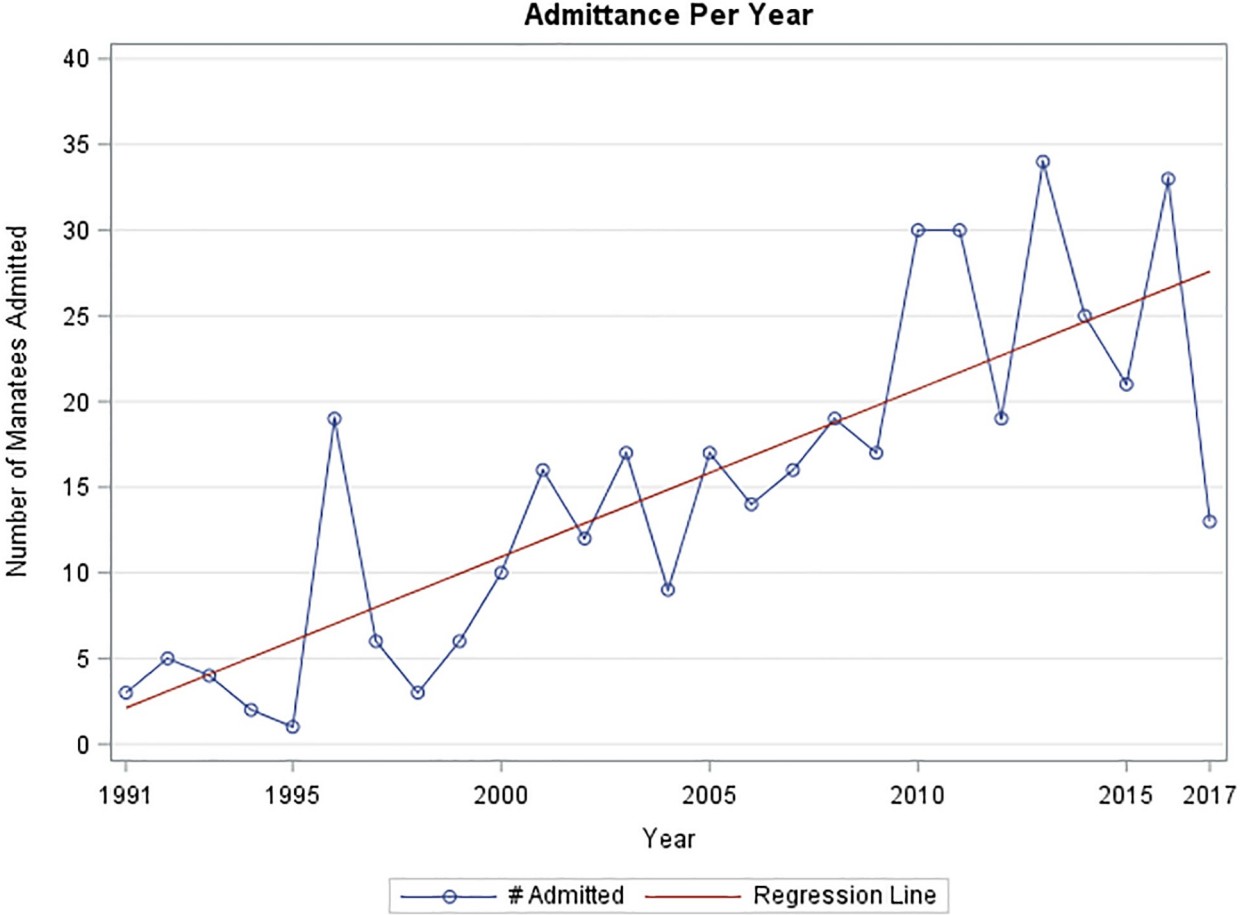

**Fig 2. Admittance of manatees at ZooTampa from August 1991 through October 2017.**

Northwest (NW). The Crystal River (CR) in Citrus County is geographically within the Northwest region but due to the density of manatees in this area and the growing human population, data is recorded for this area separately. Relationships between admittance categories, gender, age class, and rescue location were determined using statistical analysis system (SAS).

## Results

Table 1 summarizes the gender and age stratification of the manatees admitted during the study period. Adults comprised 79% of the admissions with calves accounting for 21%. The vast majority of these calves were orphans, but injured calves were occasionally admitted with injuries along with their healthy dams (n = 3) as well as healthy calves with injured dams (n = 13). Females of all ages accounted for 54% of the admissions and males of all ages comprised 46% of admissions.

Fig 2 summarizes the admissions data over the course of the study. Admittance was lowest in 1995 having only 1 admission and highest in 2013 with 34 manatees admitted. Fig 3 demonstrates the percentage of admissions by cause. Within the admittance categories for the entire study period, watercraft collisions were the highest with 36.16% of all admissions, followed by natural causes at 34.91%. The third highest category was orphaned calves at 12.7%. No other

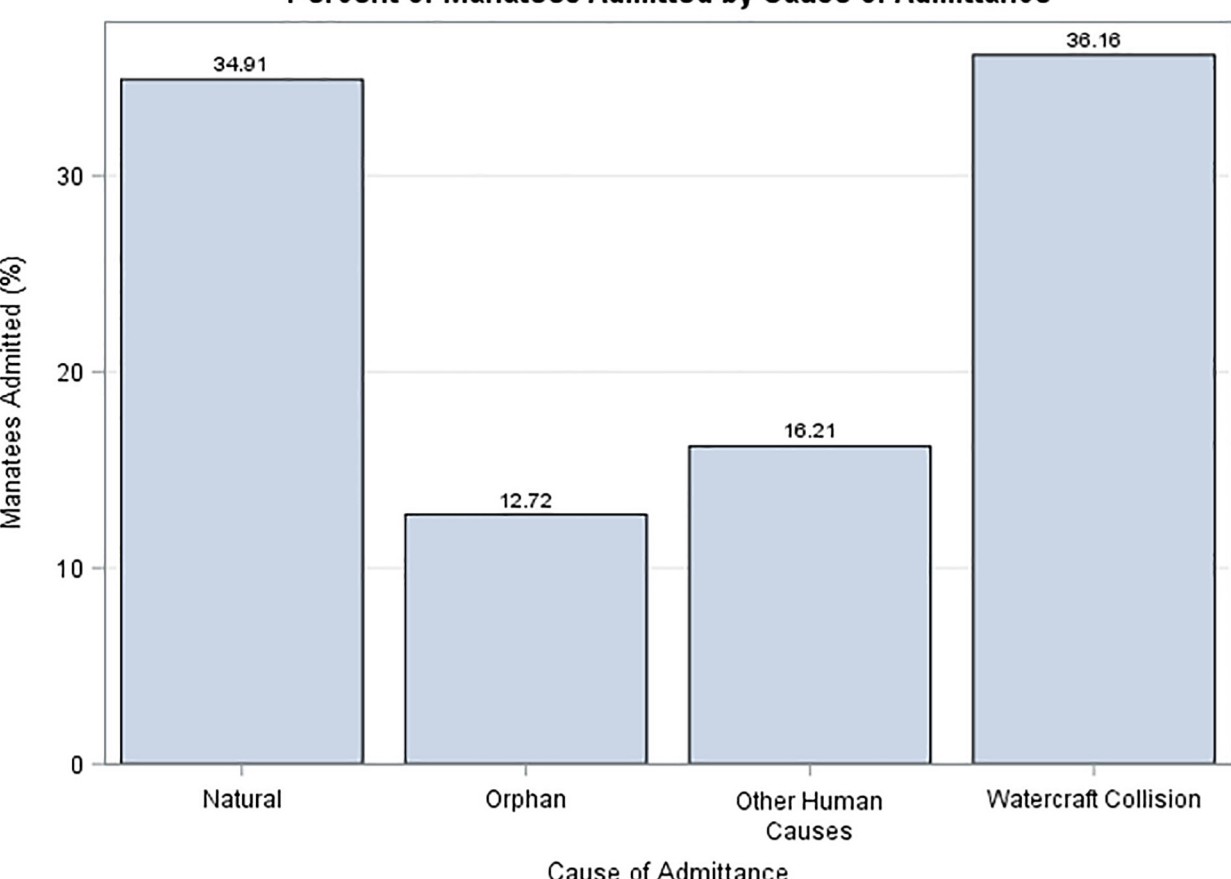

**Fig 3. Percent of manatees admitted at ZooTampa for rehabilitation by cause of admittance.**

individual categories had over 5% and they are represented here in one category, other human causes.

Fig 4 summarizes the trends in cause of admittance in 5-year time blocks over the study period. The last period, starting in 2011, was extended to the end of the study period when the rehabilitation center was closed. During this period an important change in the cause of admissions was noted with watercraft collisions becoming the dominant cause of admissions. In addition to an overall increase in admissions, each cause of admittance also tended to increase over the study period. Fig 5 demonstrates monthly admissions for the same time periods as in Fig 4. A consistent seasonal variation in total admittance was noted with the highest rates of admittance from January through April. Fig 6 demonstrates that there was also seasonal variation in peaks of admittance due to natural causes in the months of December through April, watercraft collisions May through August, and watercraft again October thru November. Admission of orphaned calves peaks in September and again in December. Orphaned calves comprise 45% of all the admitted calves.

Table 2 details the geographical demarcation of the recovery units and the associated percentages of manatee admissions from each region. Rescues and subsequent admissions primarily come from the coast of the Gulf of Mexico and related waterways. More specifically, the southwest and northwest coasts of Florida, including the Crystal River area, comprise just over 90% of all admissions.

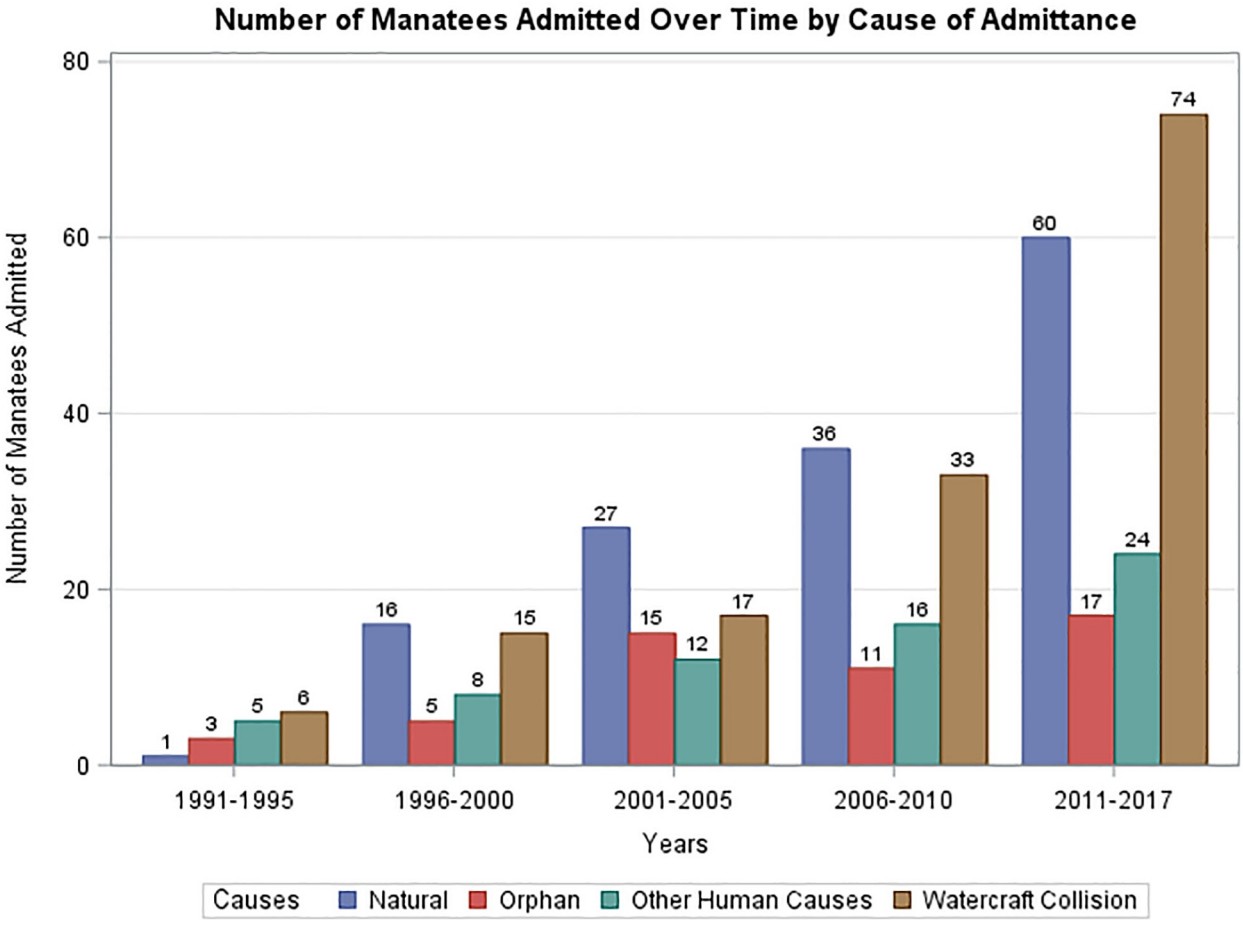

**Fig 4. Manatees admitted to ZooTampa over time by cause of admittance.**

The two dominant causes of admission, natural and watercraft were then compared to each other and are summarized in Table 3.

## Discussion

The Florida manatee population has climbed from 1,267 in 1991 to 6,620 in 2017 and estimates are now as high as 10,280 [9]. This population recovery will inevitably lead to more human-manatee conflicts and it is expected to be in the form of watercraft collisions as noted in the most recent period of this study. Watercraft collisions have played a role in admittance rates in every year of the analysis. Natural causes didn't start having an impact until 1996 when a significant mortality event due to blooms of the dinoflagellate *Karenia brevis*, which resulted in brevetoxicosis [10]. Natural causes from that point forward continued to increase as a cause for admittance and include several significant cold stress related events. The shift in admissions from natural causes to watercraft noted in the recent years may reflect this increased human-manatee interaction. Even with varied admission rates on an annual basis, the overall trend in admissions, both total and cause-specific, has been rising. The sustained trend in rising admissions over this extended period suggests a continued rise in admissions in the future.

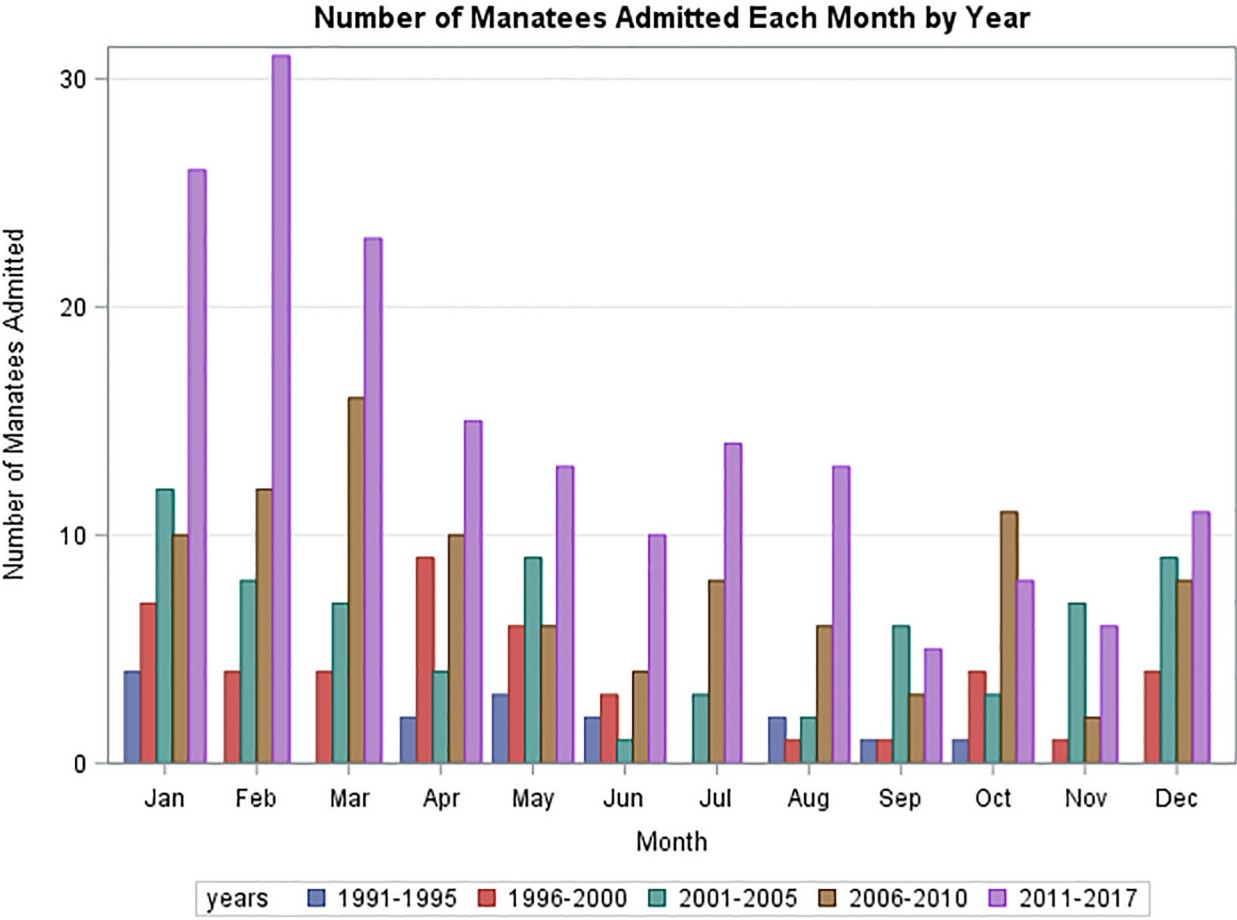

**Fig 5. Seasonal variation in admittance of manatees to ZooTampa by month over the study period.** January through April tend to be peak period for admissions.

The majority of those presented for rehabilitation at ZooTampa come primarily from the west coast, specifically southwest, which isn't surprising given the location of the facility. The northwest rescues did increase around the year 1999. This area includes the Crystal River system with a dense population of manatees. Possible causes could include increased use of an environmental resource, warm water, and/or a larger surveillance of this area. There was no significant difference in the measured parameters when comparing the two most dominant admission categories, natural and watercraft, to each other as seen in Table 3. No attributable risk could be assigned to any measured parameter when comparing the two causes of admissions.

The overall goal of obtaining and analyzing this data was to identify any trends in the admissions to better facilitate management practices to increase rehabilitation recovery rates. Confirming any seasonal tendencies in admission categories has practical applications in the resource management of the rehabilitation hospital and can reinforce public awareness campaigns regarding causes of human-related harm to manatees. Comparisons to all reported manatee rescue events with subsequent admissions for rehabilitation by the various wildlife agencies could prove the admissions data to serve as a useful proxy in the future in the event that manatees continue to be down-listed and rescue recovery efforts are streamlined or if the

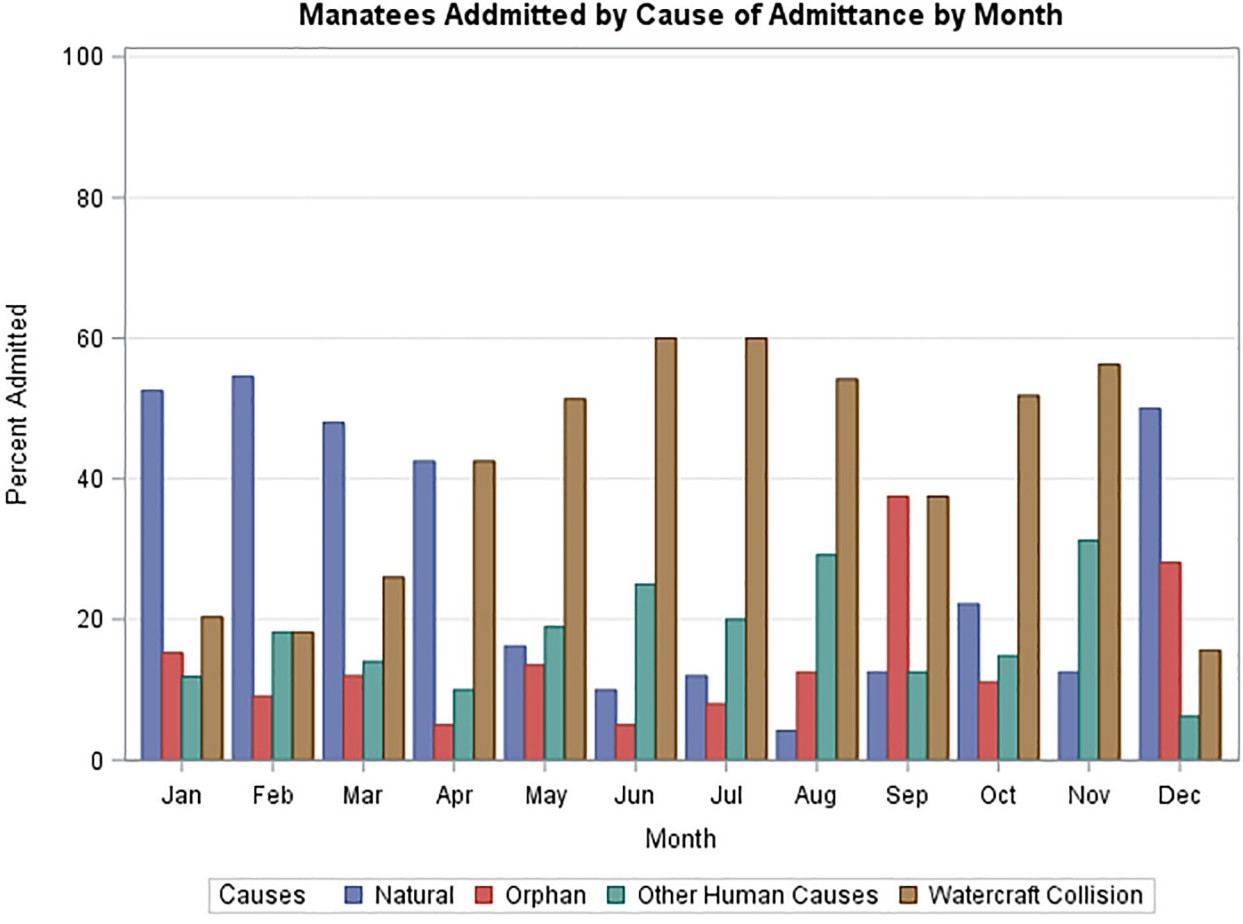

**Fig 6. Total manatees admitted by cause summated by month for entire study period.**

carcass salvage program is determined to no longer be valuable. The humanitarian effort could continue manatee rescue and rehabilitation, with its associate data collection replacing the data derived from the carcass salvage program. The data integrity could prove to be equivalent. Wildlife rehabilitation can serve as sentinels of wildlife health [11] and perhaps this eventually could be the sentinel method for Florida manatees.

**Table 2. Number of manatees admitted to ZooTampa by region over the study period.**

| Location of Rescue | No. of manatees | % of total |
|---|---|---|
| Northwest | 127 | 31.7 |
| Southwest | 215 | 53.7 |
| East Coast | 18 | 4.4 |
| Northeast | 10 | 2.4 |
| Southeast | 6 | 1.5 |
| Crystal River | 21 | 5.3 |
| Captive Born | 4 | 1.0 |
| Total | 401 | 100 |

**Table 3. Comparison between age, gender, location, and 5-year periods between manatees admitted due to natural cause vs watercraft.**

| Variable | Odds Ratio | 95%CI | | p-value |
|---|---|---|---|---|
| Adult vs. Calf | 1.26 | 0.29 | 5.53 | 0.76 |
| Female vs. Male | 1.78 | 0.98 | 3.24 | 0.06 |
| Location of Rescue | | | | |
| CR vs. SW | 0.67 | 0.14 | 3.28 | 0.63 |
| EC vs. SW | 0.20 | 0.03 | 1.53 | 0.13 |
| NE vs. SW | 0.28 | 0.04 | 2.07 | 0.22 |
| NW vs. SW | 1.20 | 0.42 | 3.46 | 0.73 |
| SE vs. SW | 1.29 | 0.08 | 21.21 | 0.86 |
| Years | | | | |
| 1991–1995 vs. 2011–2017 | 3.12 | 0.25 | 38.83 | 0.38 |
| 1996–2000 vs. 2011–2017 | 0.78 | 0.26 | 2.27 | 0.64 |
| 2001–2005 vs. 2011–2017 | 0.49 | 0.19 | 1.24 | 0.13 |
| 2006–2010 vs. 2011–2017 | 0.90 | 0.39 | 2.07 | 0.80 |

NW = Northwest: SW = Southwest; EC = East Coast; NE = Northeast; SE = Southeast; CR = Crystal River; CB = Captive Born

This initial analysis serves as a baseline and template for future analysis. Examining outcomes is the next step in an overall evaluation of rehabilitation efforts in manatees at a single facility. Mortality data, as well as details of hospitalization time, are also crucial to analyze in any rehabilitation program. Additional recommendations based on medical findings from admissions such as physical examination, hematology and serum biochemistry, and ancillary diagnostics can provide better point of care potential. This could lead to both diagnosis and even therapy on site of rescue, reducing or even eliminating some hospital admissions. Several specific causes of admissions, such as cold stress syndrome [12, 13] and brevetoxicosis [14], have some initial data analysis with recommendations being made regarding new potential therapies for both syndromes. As a result of this work, atropine has been recommended for use in brevetoxicosis [14] and anticoagulants for cold stress syndrome [13]. Combining data sets from all the qualified manatee rehabilitation centers could add insight to trends noted here or point out any potential regional differences in terms of admission patterns.

Manatees are also of concern in all parts of their natural range and conservation efforts are in place in various degrees in each location. Brazil has a significant rescue and rehabilitation effort in place and has also reported on the long-term efforts. In the Northeastern Aquatic Mammal Stranding Network's territory of Brazils Atlantic coast, an average of three Antillean manatees (*Trichechus manatus manatus*) per year have been rescued from 1987 to 2015 for a total of 77 animals [15]. The mean straight length of these alive recused manatees was 136cm and hence calves. While calves are a significant proportion of manatees rescued in the present study on the Gulf of Mexico coast of Florida, it is vastly different than the scenario described in Brazil. The main threat to this region of Brazil is associated with fisheries and calves becoming stranded [15].

Several intriguing questions are raised that are outside the intended scope of this preliminary work that deserve attention. The rise in watercraft collisions are obvious but is this the result of the manatee population recovering, a dramatic increase in the number of boats on Florida waterways, or perhaps some combination of both. This situation does have a parallel with the situation involving Florida panthers (*Puma concolor coryi*) and the increase of road killed panthers as the population in total grows. Corroborating the results of the carcass

recovery program against this study is a logical step that should be undertaken but again outside the intended scope of this preliminary work. Outcomes, release or death, are the natural progression to this review and their possible impact to the conservation status of manatees should be explored both within this study population and ideally in all manatee populations worldwide.

## Supporting information

**S1 Data.**
(SAS7BDAT)

## Acknowledgments

Numerous veterinarians, veterinary technicians, and animal care staff have been involved with the success of the manatee rehabilitation program throughout its history at ZooTampa. We are also grateful for the efforts of the veterinary nursing staff of Heather Henry, Michelle Devlin, and Ryan O'Shea.

## Author Contributions

**Conceptualization:** Ray L. Ball.

**Formal analysis:** Markuu Malmi, Janice Zgibor.

**Investigation:** Ray L. Ball.

**Project administration:** Ray L. Ball.

**Supervision:** Ray L. Ball.

**Writing – original draft:** Ray L. Ball.

**Writing – review & editing:** Markuu Malmi, Janice Zgibor.

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
