## [Decision Letter · Decision Letter 0]

6 Jan 2020

PONE-D-19-25939

Trends of the Florida manatee (Trichechus manatus latirostris) rehabilitation admissions 1991-2017.

PLOS ONE

Dear Dr. Ball,

Thank you for submitting your manuscript to PLOS ONE. After careful consideration, we feel that it has merit but does not fully meet PLOS ONE’s publication criteria as it currently stands. Therefore, we invite you to submit a revised version of the manuscript that addresses the points raised during the review process.

I have read with interest this MS and agree with the reviewers that this is an important MS with important information for the conservation of Manatees, which are threatened worldwide. Please address all comments satisfactorily, especially the questions by reviewer 1 and 3, while integrating those of reviewer 2 on statistics. Also put the MS into a wider context, drawing examples from other regions. I am looking forward to reading a revised version. 

We would appreciate receiving your revised manuscript by Feb 20 2020 11:59PM. To enhance the reproducibility of your results, we recommend that if applicable you deposit your laboratory protocols in protocols.io, where a protocol can be assigned its own identifier (DOI) such that it can be cited independently in the future. For instructions see: http://journals.plos.org/plosone/s/submission-guidelines#loc-laboratory-protocols

We look forward to receiving your revised manuscript.

Kind regards,

Ismael Aaron Kimirei, Ph.D.

Academic Editor

PLOS ONE

Journal Requirements:

"The McCune and McCann Foundations were instrumental in providing funding for both clinical fellowships and resources to facilitate diagnostic investigations."

Please provide an amended Funding Statement that declares *all* the funding or sources of support received during this specific study (whether external or internal to your organization) as detailed online in our guide for authors at http://journals.plos.org/plosone/s/submit-nowPlease state what role the funders took in the study.  If any authors received a salary from any of your funders, please state which authors and which funder. If the funders had no role, please state: "The funders had no role in study design, data collection and analysis, decision to publish, or preparation of the manuscript."

3. We note that Figure 1 in your submission contains a map image which may be copyrighted.

b.    You may seek permission from the original copyright holder of Figure 1 to publish the content specifically under the CC BY 4.0 license.

Reviewers' comments:

Reviewer's Responses to Questions

**Comments to the Author**

1. Is the manuscript technically sound, and do the data support the conclusions?

Reviewer #1: Yes

Reviewer #2: Partly

Reviewer #3: Yes

2. Has the statistical analysis been performed appropriately and rigorously? 

Reviewer #1: Yes

Reviewer #2: No

Reviewer #3: Yes

3. Have the authors made all data underlying the findings in their manuscript fully available?

Reviewer #1: No

Reviewer #2: No

Reviewer #3: Yes

4. Is the manuscript presented in an intelligible fashion and written in standard English?

Reviewer #1: Yes

Reviewer #2: Yes

Reviewer #3: Yes

5. Review Comments to the Author

Reviewer #1: In the paper “Trends of the Florida manatee (Trichechus manatus latirostris) rehabilitation admissions 1991-2017”, the authors use basic statistics to describe the trends in rehabilitation admissions of Florida manatee during almost three decades. Private zoo’s and aquariums play an important role in sirenians’ rescue and rehabilitation around the world, but there are few attempts to understand the impact of these actions on manatee conservation. This paper is very valuable to measure the role of one of these facilities to preserve Florida manatees, and can be replicated for others facilities of the same nature. This paper also shows that manatee rescue is relevant as a way to monitor threats on manatees along the time, which is very important for manatee management, specially regarding anthropogenic risks. Therefore, I consider this paper suitable to be published in PlosOne. However, the manuscript needs improvements before being considered as publishable, as following:

Materials and Methods

- The authors need to be clearer in the categorization of causes of admission. Please consider including a table separating and clearly describing each of them. What are the natural causes of death and how they were diagnosed as such?. What are the other human causes? Etc.

Results

- Although table 2 shows in a very general way the location of the rescues, it would be interesting to also present a map showing the origin of all the cases to understand the regional impact of the rescue actions.

- It is necessary to present the outcome of the rescue actions, what are the percentages of manatees deceased, released, kept in captivity (indefinitely or temporary at the moment of the paper elaboration etc). This will help the reader to understand the proportion of individuals that have been saved, and how rescue programs have a positive impact on manatee mortality.

- Fig 5 and 6 show almost the same information, I suggest to delete fig 5.

Discussion

- Please include a paragraph emphasizing the importance of manatee rescue centers not only for Florida, but also for other manatee populations around the world.

Reviewer #2: The manuscript has important data on Florida manatees admitted for rehabilitation for a period of almost two decades. Analysis of these data is crucially important to identify proper conservation actions for the subspecies.

Although the subject importance, major revision is necessary to improve the manuscript. First, the authors should provide proper statistical analyses, so adequate comparisons can be performed. Second, important improvements are necessary in Discussion section. There are no comparisons of the results obtained in this study with other obtained in Florida or other countries. I strongly recommend to the author to include in this manuscript data on the outcomes.

Reviewer #3: Dear Author,

The article "Trends in Florida Manatee (Trichechus manatus latirostris) rehabilitation admissions 1991-2017" is well-structured and presents interesting findings, with potential to support the rescue and rehabilitation program outcomes analysis. The results and discussion are clearly presented and are supported by available data. The article is well written and easy to understand.

The statistical analyses have been carried out in an appropriate manner. However, the results presented in table 3 could be discussed even if no significant correlation was found. Also, an inconsistency was identified between the data presented in rows 124 and 125 and in figure 2 (percentage of natural cause admissions and watercraft collisions). The information needs to be corrected.

Its also important to discuss if the findings corroborate the results of the carcass recovery program. Are the percentages of admission categories close to those seen in the carcass recovery? Or the rehabilitation program is being able to reduce mortality in any specific categories?

As contributions to improve the conservation outcome of the paper, I would suggest that the author's try to answer two questions: (i) is there a relation between admission categories and mortality in the rehabilitation facility?;(ii) are the number and frequency of use of watercraft increasing through the time or the trends in admission are only related to the manatee population growth?

6. PLOS authors have the option to publish the peer review history of their article (what does this mean?). If published, this will include your full peer review and any attached files.

Reviewer #1: Yes: Delma Nataly Castelblanco-Martínez

Reviewer #2: No

Reviewer #3: No

---

## [Author Response · Author response to Decision Letter 0]

21 Feb 2020

Responses to reviewers for Trends of the Florida manatee (Trichechus manatus latirostris) rehabilitation admissions 1991-2017 by Ball, Malmi, and Zgibor.

We would like to thank all the reviewers for their efforts in improving our contribution. We have attempted to address all the issues but do have some differences in opinion in what message we are looking to share at this point. These will be detailed below. The first point is that the statistical analysis has been conducted from an epidemiological perspective by epidemiologist in human public health and we believe they are rigorous and yet descriptive enough to be read and understood by readers of various academic backgrounds. With all due respect, we have elected to not alter these methods for this publication.

As for the funding statement, we were simply trying to acknowledge supporters of the entire rehabilitation program but no specific funding was obtained in the review and preparation of this manuscript. "The funders had no role in study design, data collection and analysis, decision to publish, or preparation of the manuscript."

The causes of admission and death of manatees are defined, as now noted in the manuscript, by the Florida Fish and Wildlife Conservation Commission. In our opinion this reads well and is the vernacular that is used both within our community and with the public at large when communicating about manatees. We are not sure a table would be useful. 

A detailed map showing all the rescue points would indeed be interesting but we believe too detailed for the purposes intended here. Work of this nature would be most appropriate as part of a thesis. The current manuscript is not delving into outcomes of the manatee rehabilitation; that is a follow-up manuscript so we think the descriptors are adequate. It is indeed an important aspect of the entire endeavor but again we are looking to focus our current efforts to the admission and see if there are lessons we can share.

Figures 5 and 6 do look similar but the point of each is distinctly different. Figure 5, with a revised caption, highlights the overall increase in manatee admissions over time in 5 year blocks. Figure 6 highlights the seasonality of admissions based on the admission categories.

We agree that some comparison to manatee rescues outside the USA would be appropriate. Brazil is really the only other country that does this with any frequency and that is the basis of the comparison I have added. It becomes problematic to compare facilities with the USA and within Florida as each facility is a private business, both non-profit and profit making operations. Sharing of medical data can be a challenge on an open basis that would include publication. One objective of this manuscript is to start to break down that barrier. With the completion of this manuscript, perhaps other facilities will then be open to compile their own information and data can be compared then. The fact that such a review as we are undertaking has not taken place yet really speaks volumes. This point is actually mentioned in the manuscript in lines 212 - 214.

The corrections have been made on lines 124-125 and thank you for detecting that discretion. 

A paragraph has been added in the discussion regarding manatees rescues in Brazil, which is the next largest program involved with manatees. This discussion does provide a stark contrast to the situation we see in Florida and I do think this inclusion is a good improvement. 

The last points from Reviewer 3 are exactly the questions we hope this manuscript raises and we are delighted to see this response. We would like to simply pose these questions outright in the manuscript with some thoughts and leave it unanswered as an invitation or encouragement for this work to continue. 

Sincerely

Ray L Ball

---

## [Decision Letter · Decision Letter 1]

31 Mar 2020

PONE-D-19-25939R1

Trends of the Florida manatee (Trichechus manatus latirostris) rehabilitation admissions 1991-2017.

PLOS ONE

Dear Dr. Ball,

Thank you for submitting your manuscript to PLOS ONE. After careful consideration, we feel that it has merit but does not fully meet PLOS ONE’s publication criteria as it currently stands. Therefore, we invite you to submit a revised version of the manuscript that addresses the points raised during the review process.

ACADEMIC EDITOR: The manuscript has improved a lot. There are a few edits (suggested by Reviewer 3) and some minor comments on the 5-year groupings, the stats and their meaning, and including a management statement as suggested by reviewer2. Please address these or submit a rebuttal of the same. I am looking forward to reading your revised manuscript sooner.

We would appreciate receiving your revised manuscript by May 15 2020 11:59PM. To enhance the reproducibility of your results, we recommend that if applicable you deposit your laboratory protocols in protocols.io, where a protocol can be assigned its own identifier (DOI) such that it can be cited independently in the future. For instructions see: http://journals.plos.org/plosone/s/submission-guidelines#loc-laboratory-protocols

We look forward to receiving your revised manuscript.

Kind regards,

Ismael Aaron Kimirei, Ph.D.

Academic Editor

PLOS ONE

Reviewers' comments:

Reviewer's Responses to Questions

**Comments to the Author**

1. If the authors have adequately addressed your comments raised in a previous round of review and you feel that this manuscript is now acceptable for publication, you may indicate that here to bypass the “Comments to the Author” section, enter your conflict of interest statement in the “Confidential to Editor” section, and submit your "Accept" recommendation.

Reviewer #1: All comments have been addressed

Reviewer #2: (No Response)

Reviewer #3: All comments have been addressed

2. Is the manuscript technically sound, and do the data support the conclusions?

Reviewer #1: Yes

Reviewer #2: Partly

Reviewer #3: Yes

3. Has the statistical analysis been performed appropriately and rigorously? 

Reviewer #1: N/A

Reviewer #2: No

Reviewer #3: Yes

4. Have the authors made all data underlying the findings in their manuscript fully available?

Reviewer #1: Yes

Reviewer #2: Yes

Reviewer #3: Yes

5. Is the manuscript presented in an intelligible fashion and written in standard English?

Reviewer #1: Yes

Reviewer #2: Yes

Reviewer #3: Yes

6. Review Comments to the Author

Reviewer #1: I believe the authors addressed all my comments in the new version of the MS, or explained sufficiently in the responses to reviewers- I only wanted to add that there is a phrase that seems to be repeated in the same paragraph (lines 95 and 101), please check it, and delete one of them.

95 Rehabilitated orphaned manatees must obtain 200cm before being qualified for release.

101 Calves undergoing rehabilitation must be 200cm before being considered for release.

Reviewer #2: Dear authors, your manuscript has important data on Florida manatee rehabilitation admissions. This kind of information is rare on scientific journals and I would like to congratulate you all for all the effort. I uploaded a document with all my comments and suggestions.

First, I suggest you to re-write some parts of your abstract, inserting some important data obtained in your study. I also believe you need to review your 5-year blocks. The blocks did not have the same time, mainly first and the last one. The first has 53 months, the last block has 82 months, and the other blocks have 60 months. It is difficult to compare the number of admittance in these years-blocks if they are so different.

Although I understand and respect your choice to maintain the statistical analyses, I did not understand what was your goal with this analysis, and I did not understand the results of it. I am not a statistic specialist, but I believe that I should be able to understand the results of your test, but I did not. Thus, even if you choose to maintain this logistic regression analysis, I suggest you to explain it and the results. In lines 158-159 you described “The two dominant causes of admission, natural and watercraft were then compared to each other and are summarized in Table 3.” However, when I look the table I see a different approach. It looks like you are comparing variables to see if you find any variable with more risk involved in admissions, like “Do adults have more chance to be admitted than calves?” and “Do females have more chance to be admitted than males”? Am I right? If I am, how are you comparing natural and watercraft categories here?

I also would like to see some kind of recommendations in your discussion. If watercraft admissions are increasing, what kind of improvement do you need in your rehabilitation facilities? As you verified a strong seasonal variation in watercraft and natural cause admissions, what kind of preparations are necessary in each one of these periods. I believe that before watercraft season starts you need to be prepare to perform image exams, have specific medications to treat severe wounds, etc. While natural causes, treatments are different and need other approach and preparations.

Reviewer #3: Dr. Ball,

Thanks for addressing all my comments in this reviewed version of the manuscript. I believe this paper can contribute significantly to manatees conservation in the US, specially to the rehabilitation and carcass recovery program.

Please, seriously consider publishing a new paper focusing on the outcomes of the rehabilitation program.

Best regards,

7. PLOS authors have the option to publish the peer review history of their article (what does this mean?). If published, this will include your full peer review and any attached files.

Reviewer #1: No

Reviewer #2: No

Reviewer #3: Yes: Iran Campello Normande

---

## [Author Response · Author response to Decision Letter 1]

17 Jun 2020

I have specifically adjusted the notes from Reviewer 2 in the Abstract and agree with the statement about how most readers will only read this section but I have made the following exceptions:

• Line 30 regarding why the watercraft collisions and natural causes categories were combined. The point here is just to highlight the two most dominant causes and not make any statements regarding how diverse they actually are. A reader will need to explore this.

• Lines 31-33 in regards to the relative risk (how much more likely) each of the described events will occur during certain months. I choose to leave this as is simply because we have not done specific risk analysis monthly or seasonally to quantify this. There is an obvious seasonality involved with each of these admission categories and that is mentioned as well as shown in the figures. Comments have been included in the Discussion that do highlight these trends as many readers may not be familiar with seasonal changes here in Florida.

• In Table 1 we had already defined adults and calves and were only utilizing these two age groups in this study. Inclusion of the sub-adult and juvenile category is cumbersome and subject to much debate. I rigid criteria of 200cm, the length at which a manatee would be a calf or a minimum size for release is much more solid criteria.

o “A criteria of 200cm was chosen for this study as that straight length is a determinant for both rescue and release criteria.” 

• The five year groupings are arranged in this fashion as a standard convenience for the readers and the math is relatively close if you look at a division by years. Since the time period is closer to 27 years, there is not going to be any sensible equal division that is not arbitrary. To adjust for the uneven periods, the period from 2011 on is considered a single period. There is some real practical relevance to this as that is also the period that a new medical director was hired and some changes made in the care. This has been added to the manuscript to explain the division of time. This has no real bearing on the admissions, but will be explored when outcomes are analyzed. Hence another reason to separate the admissions and outcomes into two manuscripts.

• A comment has been made regarding the seasonal preparation made in anticipation of the admission load. 

• In regards to the questions on table 3, the biostatistician and epidemiologist on this work have made the following comments:

o “Part of the confusion is that it is said "Table 3. Comparison between age, gender, location, and 5-year periods between manatees admitted due to natural cause vs watercraft." Which the reviewer took to mean that if a manatee was admitted due to natural causes compared to watercraft was out primary independent variable. In fact we reduced our data set to only those admitted due to natural causes and watercraft related reasons. Then in our logistic regression we the odds ratios represent the difference in the odds for specific groups to be admitted for natural causes. Example being that the Odds Ratio for Sex which compares females to males is 1.78, thus Females had 78% greater odds of being admitted due to natural causes when compared to males. This implies males are more likely to be admitted due to watercraft related reasons. I think that title of the table is what caused the reviewer to become confused, because he seemed to expect to see how being admitted for natural causes influenced some outcome. When in fact reason for admission was the outcome in the first place.”

o I have added some of his comments to the manuscript including the above example. In the discussion this Table 3 is again referred too pointing out all of these comparison did not yield any statistically significant findings.

---

## [Editor Report · Decision Letter 2]

22 Jun 2020

Trends of the Florida manatee (Trichechus manatus latirostris) rehabilitation admissions 1991-2017.

PONE-D-19-25939R2

Dear Dr. Ball,

We’re pleased to inform you that your manuscript has been judged scientifically suitable for publication and will be formally accepted for publication once it meets all outstanding technical requirements.

Kind regards,

Ismael Aaron Kimirei, Ph.D.

Academic Editor

PLOS ONE

Additional Editor Comments (optional):

I have enjoyed reading the revised version and think the reviewers' concerns were adequately  addressed. Please check all grammatical and spelling mistakes that remain. For example, "long term" should be a one word "long-term" (LN 217)
---

## [Editor Report · Acceptance letter]

23 Jun 2020

PONE-D-19-25939R2 

Trends of the Florida manatee (*Trichechus manatus latirostris*) rehabilitation admissions 1991-2017. 

Dear Dr. Ball:

I'm pleased to inform you that your manuscript has been deemed suitable for publication in PLOS ONE. Congratulations! Your manuscript is now with our production department. 

Kind regards, 

on behalf of

Dr. Ismael Aaron Kimirei 

Academic Editor

PLOS ONE